# An Improved Stanford Method for Persistent Scatterers Applied to 3D Building Reconstruction and Monitoring

**Bo Yang** [1] , **Huaping Xu** [1,]*, **Wei Liu** [2] , **Junxiang Ge** [3], **Chunsheng Li** [1] **and Jingwen Li** [1]

1    School of Electronic and Information Engineering, Beihang University, Beijing 100191, China
2    Department of Electronic and Electrical Engineering, University of Sheffield, Sheffield S1 3JD, UK
3    School of Electronic and Information Engineering, Nanjing University of Information Science and Technology, Nanjing 210044, China
*    Correspondence: xuhuaping@buaa.edu.cn

**Abstract:** Persistent scatterers interferometric Synthetic Aperture Radar (PS-InSAR) is capable of precise topography measurement up to sub-meter scale and monitoring subtle deformation up to mm/year scale for all the radar image pixels with stable radiometric characteristics. As a representative PS-InSAR method, the Stanford Method for Persistent Scatterers (StaMPS) is widely used due to its high density of PS points for both rural and urban areas. However, when it comes to layover regions, which usually happen in urban areas, the StaMPS is limited locally. Moreover, the measurement points are greatly reduced due to the removal of adjacent PS pixels. In this paper, an improved StaMPS method, called IStaMPS, is proposed. The PS pixels are selected with high density by the improved PS selection strategy. Moreover, the topography information not provided in StaMPS can be accurately measured in IStaMPS. Based on the data acquired by TerraSAR-X/TanDEM-X over the Terminal 3 E (T3 E) site of Beijing Capital International Airport and the Chaobai River of Beijing Shunyi District, a comparison between StaMPS-retrieved results and IStaMPS-retrieved ones was performed, which demonstrated that the density of PS points detected by IStaMPS is increased by about 1.8 and 1.6 times for these two areas respectively. Through comparisons of local statistical results of topography estimation and mean deformation rate, the improvement granted by the proposed IStaMPS was demonstrated for both urban areas with complex buildings or man-made targets and non-urban areas with natural targets. In terms of the spatiotemporal deformation variation, the northwest region of T3 E experienced an exceptional uplift during the period from June 2012 to August 2015, and the maximum uplift rate is approximately 4.2 mm per year.

**Keywords:** InSAR; PS-InSAR; StaMPS; topography measurement; deformation monitoring

## 1. Introduction

Space-based technology is an efficient way to reconstruct 3D topography and monitor both natural and man-made disasters, such as earthquakes [1–3], volcanoes [4–7], landslides [8–11], and anthropogenic subsidence or uplift due to fluid withdrawal or injection [12–15]. One representative example is the spaceborne Synthetic Aperture Radar (SAR), which has a high spatial-temporal resolution and a broad spatial coverage, capable of precise topography measurement and deformation monitoring using advanced multi-temporal interferometric SAR (InSAR) techniques [16], such as persistent scatterer InSAR (PS-InSAR) [17–20], SBAS [21,22], SqueeSAR [23], CAESAR [24,25], PD-PSInSAR [26], and TomoSAR [27–29]. With its capability of measuring topography up to sub-meter scale and monitoring subtle deformation up to mm/year scale, PS-InSAR has been widely employed

in various applications of remote sensing [30], and even for phase calibration of other advanced multi-temporal InSAR techniques [31]. The key idea behind these methods lies in the identification of dominant scatterer pixels (PS), which are stable over a long acquisition time and for a wide look-angle span.

PS pixels are selected based on their phase variation in time [17–19], and there are mainly two steps in identifying PS pixels. Firstly, a set of PS candidates (PSCs) with high signal-to-noise ratio (SNR) is identified by analyzing their amplitude variation, pixel by pixel, in a series of interferograms. Secondly, a deformation model related to the temporal variation, such as a linear or periodic model, is assumed for the estimation of temporal coherence. Each pixel in PSCs is tested for phase stability by temporal coherence. Only those whose phase variation fits into the assumed deformation model are deemed stable. These methods have been very successful in detecting PS pixels and investigating topography of urban areas. Especially, the quantity of PS pixels in urban areas has further increased for the recently proposed Iterative PS-InSAR method (IPSI) [32]. For these urban areas, the ground surface is in primarily steady state or undergoes constant or periodic deformation. Moreover, there are many strong and dominated scatterers resulting from man-made objects, and, typically, the density of PS pixels exceeds 3–4 points per km$^2$ [33]. Physically, the stable and dominate scatterers might be the roofs and edges, which reflect echoes directly backwards similar to a mirror, or once from the ground and once from a perpendicular structure resulting in a double bounce [34]. However, because the unmodeled phase resulting from spatial difference would be more than 0.6 rad, leading to unreliable phase unwrapping [33], these approaches fail in two aspects. On the one hand, it fails for most natural terrains, such as most of volcanoes and the rural areas, where the scatterers with high SNR are rare. The distance of neighboring PS pixels is so large that the absolute value of unmodeled phase resulting from spatial atmosphere and deformation difference exceeds $\pi$ in phase unwrapping. On the other hand, due to the inaccurate deformation model, many stable pixels in natural areas are dropped from PSCs, which leads to a small density of PS and a large distance between neighboring PS pixels.

In [35,36], Hooper et al. proposed the so-called Stanford Method for PS (StaMPS), to extract the deformation signal over an area that contains few man-made structures. The temporal model for deformation during the observation interval does not need to be predefined. StaMPS is widely used to investigate deformation in various areas with different sensors [37–42]. An initial set of PSCs is selected based on analysis of amplitude with an adaptive threshold, and the phase stability of these pixels is analyzed based on the spatial correlation in an iterative process. Then, the PSCs are selected further with an adaptive spatial coherence threshold, which is obtained by a PS and non-PS probabilistic model. Finally, an optional selection stage is included to reject pixels that persist only in a subset of the interferograms and those that are dominated by scatterers in adjacent PS pixels. However, there are three limitations. Firstly, the probabilistic model may not be suitable for the initial PSCs set. For complex scenes, there exist layover pixels within multiple dominant scatterers. If the layover pixels, especially, overlaid by multiple scatterers with approximately equal SNR and close distance, are included in PSCs, then the probabilistic model for finding the adaptive threshold will become inaccurate. This leads to the phenomenon that some layover pixels are misidentified as PS, while many PS near layover pixels are dropped from the initial PSCs. Secondly, under the assumption that the adjacent PS pixels possess the same dominant scatterer, stable adjacent pixels of PS are mostly dropped from PSCs. The assumption is less accurate for scenes within many extended targets [42]. Thirdly, the accurate topography is not investigated in StaMPS, although the external digital elevation model (DEM) is typically of low resolution and there may exist great deviation compared with current topography.

In this paper, a modified StaMPS method called IStaMPS (Improved Stanford Method for PS) is proposed. The main improvement is focused on PS selections. Moreover, the topography information not provided in StaMPS is accurately measured in IStaMPS. To verify the improvement, both StaMPS and IStaMPS were applied to the same dataset acquired by TerraSAR-X/TanDEM-X over Terminal 3 E (T3 E) of the Beijing Capital International Airport and the Chaobai River of Beijing Shunyi District,

where there are multiple types of scattering characteristics, including man-made and natural targets. It is shown that the density of PS points detected by IStaMPS is increased by about 1.8 and 1.6 times for these two areas, respectively. The enhanced performance of IStaMPS was illustrated by each step of selected points using the T3 E site as an example. Through comparisons of local statistical results of topography estimation and mean deformation rate, the improvement granted by the proposed IStaMPS was demonstrated quantitatively. Finally, the spatiotemporal deformation variations of T3 E during the period from June 2012 to August 2015 were investigated.

## 2. Methodology

Suppose there are $M + 1$ single-look complex (SLC) SAR images, and then $M$ interferograms are generated with respect to a common master image. For the stack of interferograms, the key procedures of StaMPS and IStaMPS are shown in Figure 1, which are marked with red dotted box and red solid box, respectively. The main difference of PS identification is the selection of initial PSCs and the removal of unstable adjacent PS pixels. Initially, a subset of pixels is selected based on analysis of the mean amplitude (MA) and the amplitude dispersion index (ADI) of SAR stack. Then, the reason the initial set of PSCs is more suitable than that of StaMPS for the PS probabilistic model, is explained in detail. Moreover, an optional selection is included to reject adjacent pixels with low temporal coherence. The difference is detailed in the following subsections. In addition, to accurately retrieve both topography and deformation, there are several aspects that differ from StaMPS in the procedure of results retrieving. Firstly, the spatial-correlation residual topography phase is kept before 3D phase unwrapping. Secondly, the residual topography and linear deformation are roughly estimated and subtracted. This step ensures accurate estimation of the atmosphere and orbit phase. Finally, to remove the noise part of deformation phase, the deformation time series are obtained by spatially and temporally lowpass filtering.

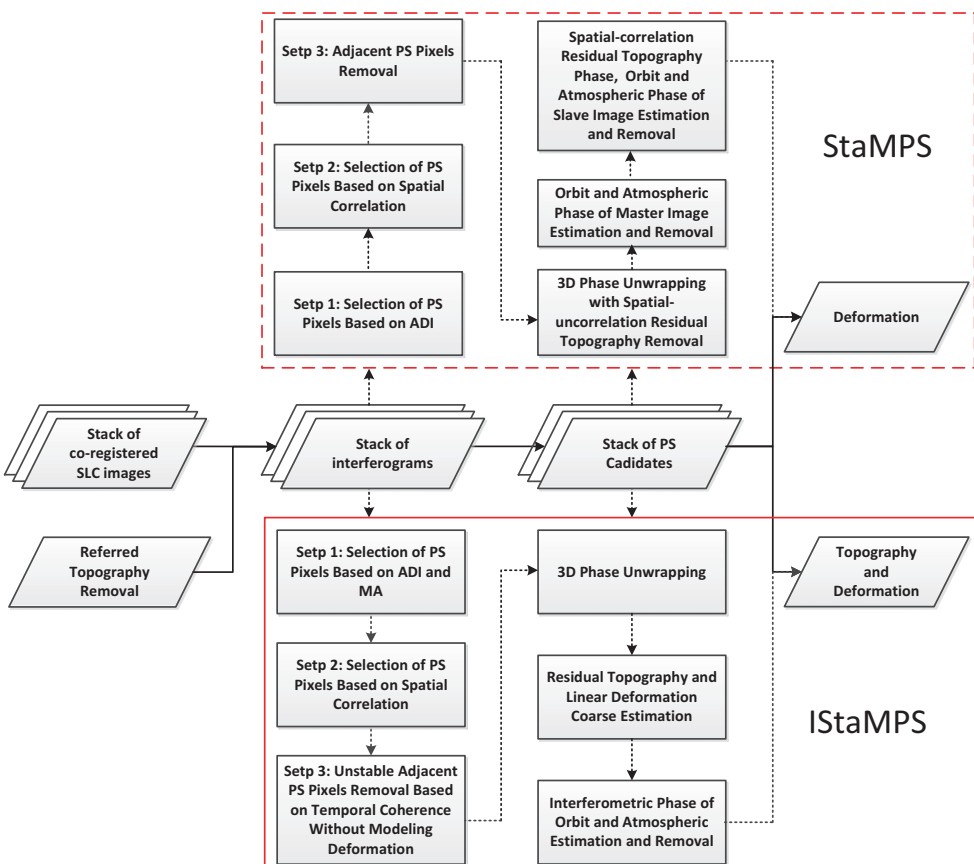

**Figure 1.** Flow diagram of the implemented method in this study.

### 2.1. Selection of PS Pixels Based on ADI and MA

When performing the phase stability analysis based on spatial correlation in StaMPS, the normalizing weighting factors of neighboring pixels are typically set with the SNR associated with brightness. In fact, the brightness of PS pixel is typically less than that of layover pixels. If layover pixels within the superposition of multiple strong scatterers are not excluded, the coherence of the PS near the layover pixels may not be exactly calculated based on spatial correlation. As illustrated in Figure 2, pixel $P_1$ associated with the resolution cell in red is mainly overlaid by the two strong dominant scatterers of building eave and tree crown. If the pixel $P_1$ is not excluded, the coherence calculation of PS pixel $P_2$ would be influenced due to the large weight of pixel $P_1$. As a consequence, the coherence of true PS points may decrease and the coherence of non-PS points may increase.

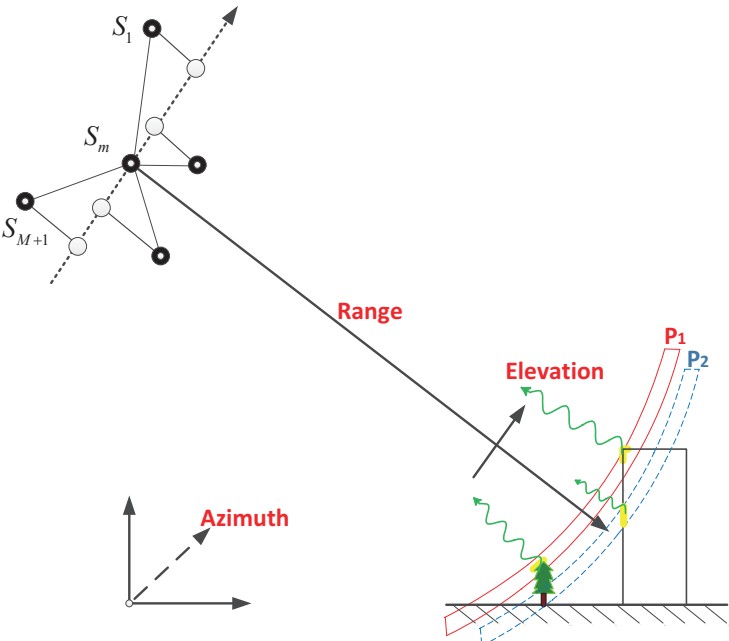

**Figure 2.** Geometric diagram of two resolution cells: $P_1$ represents a layover pixel with two strong point-like scatterers' superposition and $P_2$ represents a PS pixel within only one scatterer from building façade.

When it comes to the PS pixels selection based on spatial correlation, the population of PSCs pixels is treated as the union of two populations, population $X$ containing only PS pixels and population $Y$ containing only non-PS pixels. Then, the probability density model $p(\gamma_P)$ associated with the coherence $\gamma_P$ of pixel $P$ is a weighted sum of the probability density $p_X(\gamma_P)$ for the PS pixels and the probability density $p_Y(\gamma_P)$ for the non-PS pixels, i.e.,

$$p(\gamma_P) = (1 - \beta)p_X(\gamma_P) + \beta p_Y(\gamma_P) \tag{1}$$

where $\beta$ is the weighting factor with $0 \leq \beta \leq 1$. By binning and normalizing the values of $\gamma_P$, $p(\gamma_P)$ can be obtained through histogram, while $p_Y(\gamma_P)$ can be derived using a large number ($10^6$ used here) of simulated pseudo-pixels with random noise phase to model the noise term, and then an adaptive coherence threshold on $\gamma_P$ can be derived to identify the PS pixels. Due to adaptivity of the coherence threshold obtained based on the PS probabilistic model, it can effectively select the PS pixels, especially for the areas with few man-made targets.

However, the probabilistic model in Equation (1) may be limited for complex areas, where there exist some special layover pixels within multiple strong dominant scatterers, i.e., the pixel $P_1$ in Figure 2. Due to the partial correlation of layover pixels, the simulated random noise phase population would

not exactly contain this kind of layover population. Then, $p_Y(\gamma_P)$ simulated by random noise phase is not accurate enough to model the population $Y$ of initial PSCs, which contains the special layover pixels. As a result, the adaptive threshold derived based on the probabilistic model in Equation (1) would not be exact in selecting stable PS. Some layover pixels would be misidentified as PS, while many stable PS near the layover pixels would be dropped from the PSCs.

Accordingly, some special layover pixels within multiple strong scatterers must be removed before coherence calculation and PS pixels selection based on spatial correlation. Since there is a statistical relationship between amplitude stability and phase stability [18], the initial pixels for phase stability analysis are selected based on the statistics of their amplitudes. In this work, the non-PS pixels within an area of zero scatterer and PS pixels within an area of a single dominant scatterer are selected by thresholding on ADI and MA. The MA $\mu_{A,P}$ and ADI $D_{A,P}$ of $P$ pixel are defined as [18,43]

$$\mu_{A,P} = \frac{1}{M+1} \sum_{m=1}^{M+1} |\mathbf{S}_{m,P}| \tag{2}$$

$$D_{A,P} = \sqrt{\frac{1}{M+1} \sum_{m=1}^{M+1} |\mathbf{S}_{m,P}|^2 - \left( \frac{1}{M+1} \sum_{m=1}^{M+1} |\mathbf{S}_{m,P}| \right)^2} \Big/ \frac{1}{M+1} \sum_{m=1}^{M+1} |\mathbf{S}_{m,P}| \tag{3}$$

where $\mathbf{S}_{m,P}$ is the complex value of the $m$th observation at pixel $P$, and $|\cdot|$ is the absolute value operator. They measures the scattering brightness of a pixel and the standard deviation (std) from a series of amplitude values, respectively. Since the layover pixels have a high brightness level [43,44], thresholds on $\mu_{A,P}$ and $D_{A,P}$ ensure that most of layover pixels and non-PS pixels are excluded. Then, the initial set of PSCs ensures the correctness of coherence $\gamma_P$ estimated based on spatial correlation. Moreover, the initial PSCs are guaranteed to be suitable for the probabilistic model to further select the reliable PS. Typically, the threshold value on $D_{A,P}$ is predefined in the region of 0.4, and the threshold value on $\mu_{A,P}$ is predefined to be above $\mu_{A,85\%}$, where $\mu_{A,85\%}$ means the first 85% of pixels with their mean amplitudes sorted from smallest to largest are selected.

## 2.2. Removal of Unstable Adjacent PS Pixels

In StaMPS, to ensure the accuracy of 3D phase unwrapping, adjacent pixels of PS are entirely dropped from PSCs due to the assumption that the dominant scatterer in PS pixel appears in all other adjacent pixels. The assumption is too harsh for scenes with many extended targets.

Here, the stable adjacent PS pixels are identified based on the phase stability analysis of temporal coherence. Different from conventional temporal coherence [18,19,31], here the deformation for temporal evolution is not modeled, which is motivated from the 3D phase unwrapping theory [45].

Mathematically, the wrapped phase, $W\{\Delta\phi_{P,i}\}$, of the pixel $P$ of the $i$th differential interferogram can be written as the wrapped sum of five terms [35]

$$W\{\Delta\phi_{P,i}\} = W\{\Delta\phi_{H,P,i} + \phi_{D,P,i} + \phi_{A,P,i} + \phi_{O,P,i} + \phi_{N,P,i}\} \tag{4}$$

where $\Delta\phi_{H,P,i} = \frac{4\pi b_{\perp,i}\Delta h}{\lambda r \sin\theta}$ is the residual topographic phase due to the coarse DEM; $\phi_{D,P,i}$ is the phase change due to movement of the scatterer along the satellite Line of Sight (LOS) direction; $\phi_{A,P,i}$ is the phase due to the difference in atmospheric delay between passes; $\phi_{O,P,i}$ is the residual phase due to satellite orbit inaccuracies; $\phi_{N,P,i}$ is a noise term due to the scattering variability, thermal noise, co-registration errors and position uncertainty of the phase center in azimuth; and $W\{\cdot\}$ denotes the wrapping operator.

Without assuming a particular deformation model for temporal evolution, each PSC phase is filtered by a low-pass temporal operator. After subtracting the resulting filtered phase value $\Delta\phi_{P,i}^{tl}$ from $\Delta\phi_{P,i}$, the rewrapping phase gives

$$W\left\{\Delta\phi_{P,i} - \Delta\phi_{P,i}^{tl}\right\} = W\left\{\Delta\phi_{H,P,i}^{th} + \phi_{D,P,i}^{th} + \phi_{A,P,i}^{th} + \phi_{O,P,i}^{th} + \phi_{N,P,i}\right\} \tag{5}$$

where $\phi^{th}$ denotes the temporally uncorrelated part of $\phi$. $\phi_{D,P,i}^{th}$ is very small and can be considered as zero. For spatial phase unwrapping, the Delaunay triangulation network is used to connect the sparse PSCs via spatial arcs [17]. Since the temporal high-pass part of $\phi_{N,P,i}$ does not change, the double-difference wrapped phase of the $P_1$-$P_2$ arc after removal of temporal low-pass filtering is given by

$$W\left\{\Delta_{P_1}^{P_2}\Delta\phi_{P,i} - \Delta_{P_1}^{P_2}\Delta\phi_{P,i}^{tl}\right\} = W\left\{\Delta_{P_1}^{P_2}\Delta\phi_{H,P,i}^{th} + \Delta_{P_1}^{P_2}\phi_{D,P,i}^{th} + \Delta_{P_1}^{P_2}\phi_{A,P,i}^{th} + \Delta_{P_1}^{P_2}\phi_{O,P,i}^{th} + \Delta_{P_1}^{P_2}\phi_{N,P,i}\right\} \quad (6)$$

where $\Delta_{P_1}^{P_2}$ represents the spatial difference operator between pixels $P_1$ and $P_2$. Due to the spatial correlation of $\phi_{A,P,i}$ and $\phi_{O,P,i}$, the spatial difference operator of adjacent pixels mitigates the effect of atmosphere and orbit, and thus $\Delta_{P_1}^{P_2}\phi_{A,P,i}^{th} \approx \Delta_{P_1}^{P_2}\phi_{O,P,i}^{th} \approx 0$. Due to the spatial and temporal correlation of $\phi_{D,P,i}$, $\Delta_{P_1}^{P_2}\phi_{D,P,i}^{th}$ is very small and can be considered as zero. Afterwards, the temporal high-pass part of differential topography $\Delta_{P_1}^{P_2}\Delta h_p^{th}$ is estimated by maximizing the following temporal coherence measurement

$$\hat{\gamma}_{P_1 P_2} = \arg\max_{\Delta h_p^{th}} \left| \frac{1}{M} \sum_{i=1}^{M} \exp\left[j\left(W\{\Delta_{P_1}^{P_2}\Delta\phi_{P,i} - \Delta_{P_1}^{P_2}\Delta\phi_{P,i}^{tl}\} - \frac{4\pi b_{\perp,i}\Delta_{P_1}^{P_2}\Delta h_p^{th}}{\lambda r \sin\theta_p}\right)\right] \right| \quad (7)$$

　　Then, the noise level and the temporal coherence of the $P_1$-$P_2$ arc are obtained. Furthermore, the noise level of each PSC is estimated by the integration of differential noise of arcs. Finally, the temporal coherence of each PSC is obtained. The pixel whose coherence exceeds a fixed threshold (typically, 0.6–0.7), is selected to be the most likely PS. After dropping the pixels with low temporal coherence, a new Delaunay triangulation network is formed and $\hat{\phi}_{N,P,i}$ is reestimated. Several iterations (3–5 loops) are implemented to refine the selection of PS pixels.

*2.3. Topography and Deformation Estimation*

2.3.1. 3D Phase Unwrapping

　　Once the PS pixels are selected, the original wrapped interferogram phase must be unwrapped firstly. The 3D phase unwrapping method [45], which can effectively mitigate the phase jump of PS data in spatial and temporal dimensions, is adopted. There are four steps in its implementation. Firstly, resample the wrapped phase to grids and spatially low-pass filter the phase. Secondly, unwrap the phase difference of nearby grids temporally. Thirdly, unwrap phase spatially using maximum a posteriori (MAP) cost functions. Finally, interpolate 3D PS phase from unwrapped gridded interferograms. The results of unwrapping can be summarized as

$$\Delta\phi_{P,i} = W\{\Delta\phi_{P,i}\} + 2k_{P,i}\pi = \Delta\phi_{H,P,i} + \phi_{D,P,i} + \phi_{A,P,i} + \phi_{O,P,i} + \phi_{N,P,i} \quad (8)$$

2.3.2. Coarse Estimation of Topography Error and Deformation Rate

　　The temporal difference of unwrapped phase in Equation (8) is given by

$$\Delta_{i_1}^{i_2}\Delta\phi_{P,i} = \Delta_{i_1}^{i_2}\Delta\phi_{H,P,i} + \Delta_{i_1}^{i_2}\phi_{D,P,i} + \Delta_{i_1}^{i_2}\phi_{A,P,i} + \Delta_{i_1}^{i_2}\phi_{O,P,i} + \Delta_{i_1}^{i_2}\phi_{N,P,i} \quad (9)$$

where $\phi_{D,P,i}$ can be decomposed into linear and nonlinear components $\phi_{DM,P,i}$ and $\phi_{DN,P,i}$ [19]. The linear component is the dominant part for deformation and modeled by $\phi_{DM,P,i} = 4\pi t_i v/\lambda$, where $v$ is the mean deformation rate. The residual height $\Delta\hat{h}$ and linear deformation rate $\hat{v}$ are roughly estimated by the least square method, and then $\Delta\hat{\phi}_{H,P,i}$ and $\hat{\phi}_{DM,P,i}$ are obtained.

### 2.3.3. Orbit Phase Estimation

After removal of the retrieved topographic phase and linear deformation phase, the unwrapped phase is written by

$$\Delta\phi_{P,i} - \Delta\hat{\phi}_{H,P,i} - \hat{\phi}_{DM,P,i} = \phi_{DN,P,i} + \phi_{A,P,i} + \phi_{O,P,i} + \delta_{N,P,i} \tag{10}$$

In fact, the orbit phase of each interferogram can be linearly modeled associated with the spatial position [19,46], that is,

$$\phi_{O,P,i} = a_i \cdot P_x + b_i \cdot P_y + c_i \tag{11}$$

where $(P_x, P_y)$ denotes the SAR coordinate position of $P$, and $a_i, b_i$ and $c_i$ are the linear model parameters. Since the nonlinear deformation and atmosphere phase are small and not linearly modeled by Equation (11), $a_i, b_i$ and $c_i$ are retrieved by 2D linear regression analysis, and then $\hat{\phi}_{O,P,i}$ is obtained.

### 2.3.4. Atmospheric Phase Estimation

After removal of the retrieved topographic and mean deformation phase as well as orbit phase, the unwrapped phase becomes

$$\Delta\psi_{P,i} = \Delta\phi_{P,i} - \Delta\hat{\phi}_{H,P,i} - \hat{\phi}_{DM,P,i} - \hat{\phi}_{O,P,i} \tag{12}$$

Based on the different spatial and temporal behavior of atmospheric phase and nonlinear deformation phase, the atmospheric phase can be separated from deformation through temporal band-pass filtering [17]. Firstly, a temporally low-pass filter in time $\mathcal{L}_i^{tl}$ is applied to the phase difference between neighboring PS pixels. Then, the separated high-pass differential phase is used to estimate atmospheric phase $\phi_{A,P,i}$ by

$$\hat{\phi}_{A,P,i} = \left[\Delta_{P_1}^{P_2}\right]^{-1}\left\{\Delta_{P_1}^{P_2}\Delta\psi_{P,i} - \mathcal{L}_i^{tl}\left\{\Delta_{P_1}^{P_2}\Delta\psi_{P,i}\right\}\right\} \tag{13}$$

where residual noise is suppressed during the least square inversion $\left[\Delta_{P_1}^{P_2}\right]^{-1}$ of differential atmospheric phase.

### 2.3.5. Topography and Deformation Time Series

Finally, after removal of atmospheric and orbit phase, an iterative regression analysis is performed to improve the model parameters including the residual topography, the linear deformation rate and the deformation time series. Mathematically, the calibrated phase is given by

$$\Delta\phi_{P,i} - \hat{\phi}_{O,P,i} - \hat{\phi}_{A,P,i} = \Delta\phi_{H,P,i} + \phi_{DM,P,i} + \phi_{DN,P,i} + \delta_{N,P,i} \tag{14}$$

According to Equation (9), topography error and deformation rate are estimated. Moreover, the deformation time series is obtained by

$$\hat{\phi}_{D,P,i} \approx \mathcal{L}_P^{sl}\left\{\mathcal{L}_i^{tl}\left\{[\Delta\phi_{P,i} - \hat{\phi}_{O,P,i} - \hat{\phi}_{A,P,i} - \Delta\hat{\phi}_{H,P,i}]\right\}\right\} \tag{15}$$

where spatially and temporally low-pass filters in [35] are adopted to reduce the noise phase.

## 3. Study Area and Dataset Used

Shunyi District is located at the northeast of Beijing, China. There are multiple types of man-made and natural targets for urban and non-urban areas study, such as the Beijing Capital International Airport and the Chaobai River. As shown in Figure 3, there are three terminals (T1, T2, and T3) and three runways numbered 36L, 36R, and 01 at the airport. T3 includes three buildings, T3 C, T3 D and

T3 E. T3 and runway 01 were built in 2008 to cope with the extra traffic brought by the 2008 Olympic visitors. The Chaobai River is surrounded by several villages, and crossed by Baima Road and near Youdi Road.

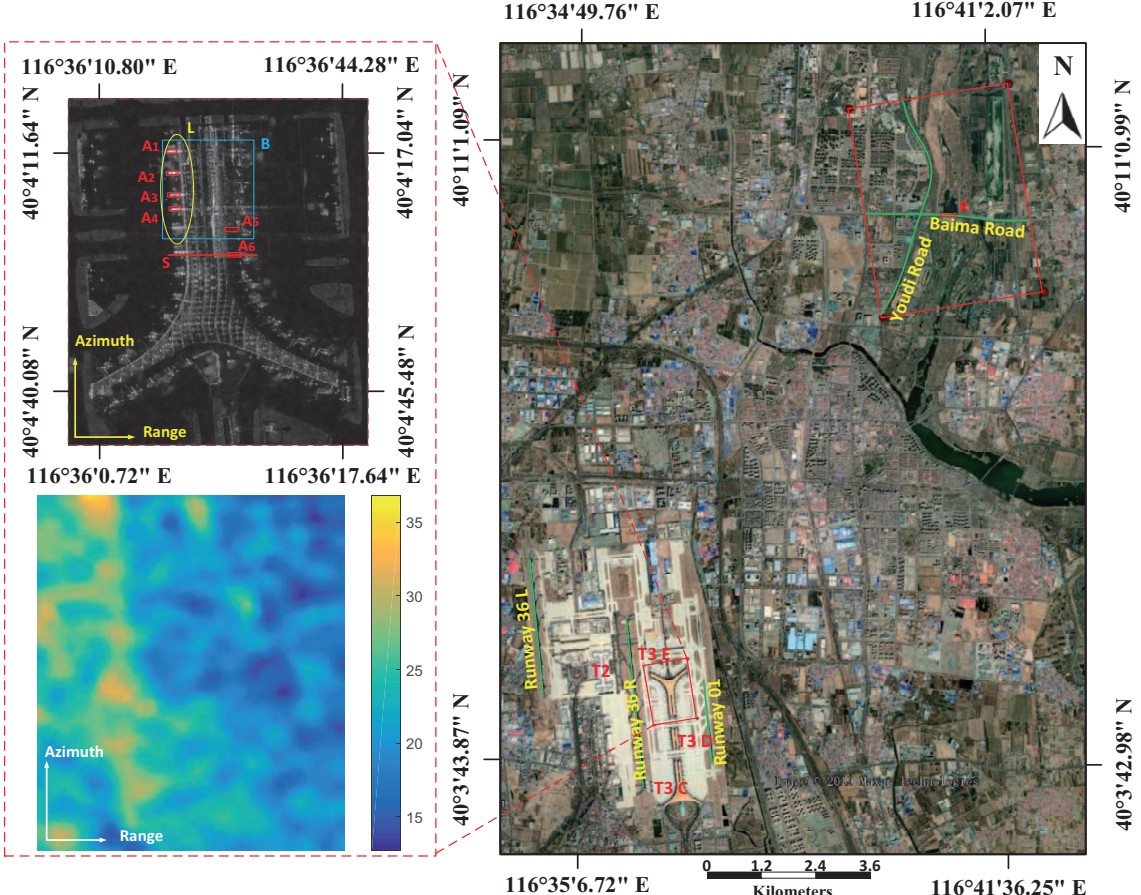

**Figure 3.** Shunyi District from Google Earth. The study areas are the T3 E building and the Chaobai River marked with red rectangles. T he top and bottom sub-images at the left side are SAR intensity map and referred DEM map of T3 E, respectively. Several lounge bridges marked with $A_1$–$A_6$ and most likely overlaid area $L$ and part of the T3 E building marked by $B$ are selected for quantitative comparisons of urban areas. The Chaobai River bridge marked with region $R$ is selected for quantitative comparisons of non-urban areas.

To assess the performance of the proposed IStaMPS, the T3 E and Chaobai River sites were selected as the study areas, which are marked with red box in Figure 3. On the one hand, there are multiple types of scattering characteristics over the T3 E site, including complex building and runways, which is adequate to assess the superiority of IStaMPS over StaMPS for urban areas. Due to the layover effect taking place in the building, this is a relatively challenging area for both algorithms. One SAR intensity image observed in this area is shown at the top-left side of Figure 3. The roof of the T3 E building has a streamlined shape. On the other hand, there are many bare soils and rocks located on both sides of the river. They are adequate to validate the reliability of IStaMPS for non-urban areas.

The dataset used in this study consists of a stack of 25 TerraSAR-X/TanDEM-X images, covering ground areas of 1.3 km × 1 km and 4.6 km × 3.6 km, across which the T3 E building and the Chaobai River are located, respectively. The stack of 25 TerraSAR-X/TanDEM-X images were acquired from June 2012 to August 2015 from the ascending track provided by the German Aerospace Center. The TerraSAR-X image acquired on 10 October 2013 was selected as the master image and the remaining images were jointly co-registered with it. The maximum temporal baseline is about 3.4 years,

and the perpendicular baseline ranges from $-182$ m to 312 m. The temporal/perpendicular baseline plot for the SAR dataset is shown in Figure 4. The stack was imaged in Stripmap mode with a resolution up to 3 m at HH single polarization with a mean incidence angle of 34.74° and 35.37°. Moreover, the one arc-second SRTM DEM associated with the studied areas was used to derive the referred topographic phase. For example, the left side of Figure 3 lists the referred DEM associated with the T3 E site. After removing the derived phase from interferograms, series of differential interferograms were generated. It is worth noting that the bias of residual topography is very large for the T3 E building. Thus, a 3D reconstruction was necessary for investigation.

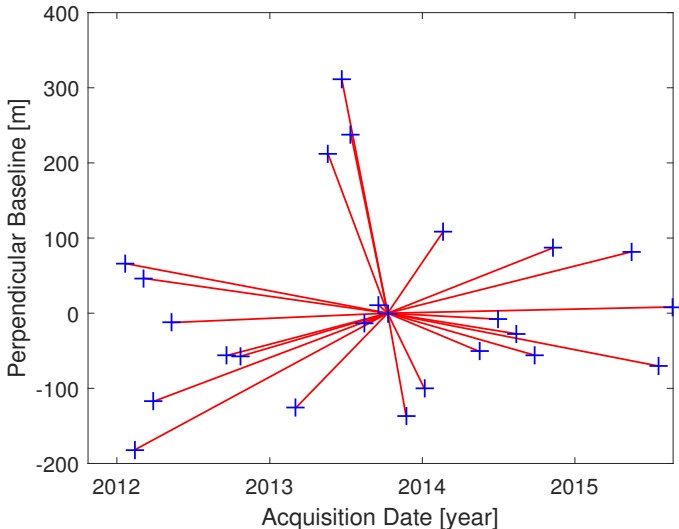

**Figure 4.** Acquisition date and perpendicular baseline plot.

## 4. Experimental Results and Discussions

The same preprocessing method was used to generate interferograms of StaMPS and IStaMPS for a fair comparison. Due to lack of accurate topography measurement in StaMPS, the estimation method presented in Section 2.3 was used to retrieve the topography and deformation information for both the studied urban and non-urban areas. All results are shown in SAR coordinate.

### 4.1. PS Pixels Selection Results

Taking the T3 E site as an example, the detailed selection process for PS pixels by the IStaMPS method is described as follows. According to the distribution histograms of $D_A$ and $\mu_A$ of the dataset shown in Figure 5, a subset of pixels most likely to be PS was selected as PSCs by thresholding the maximum value of $D_A$ and $\mu_A$ at 0.4 and 5000, respectively. This step ensured that most pixels with a single-scatterer stable phase were included and most pixels overlaid by multiple strong scatterers were excluded from the initial PSCs. Secondly, a linear model for spatial coherence threshold associated with $D_A$ was derived from the initial PSCs based on StaMPS, as shown in Figure 6a. Then, the PSCs were updated by rejecting those pixels less than the adaptive coherence. Finally, the updated PSCs were connected with the Delaunay triangulation network, as shown in Figure 6b. The pixels with temporal coherence below 0.65 were dropped according to the process described in Section 2.2.

In total, 3183 PS pixels (1153 pixels/km$^2$) and 8980 PS pixels (3253 pixels/km$^2$) over the T3 E site were detected by StaMPS and IStaMPS, respectively. The density of the reliable measurement points detected by IStaMPS is about 2.8 times that detected by StaMPS. Subsequently, a comparison of measurement points of each step identified by the two methods is presented in Figure 7, where the three-step PSCs results are shown in the top row and bottom row, respectively. For the first step of amplitude analysis, there were 47,095 PSCs initially selected by StaMPS, while 46,554 PSCs were selected by applying IStaMPS, where the smaller number of IStaMPS was due to its extra constraint

of $\mu_A$. For the second step, IStaMPS provided very good coverage over most parts of the man-made structures such as roofs and eaves of the T3E building, while StaMPS could not provide detailed information of T3 E eaves in the *L* region marked in Figure 5. Furthermore, the PSCs detection ratio, which is the number of latter PSCs over that of previous PSCs, was 0.383 for IStaMPS, higher than the value of 0.380 by StaMPS. The reason may be that the probabilistic model in Equation (1) is more suitable for the initial PSCs selected by IStaMPS than that by StaMPS. Thus, more reliable PSCs with a single dominant scatterer were effectively selected by the proposed IStaMPS. For example, some layover pixels were selected as PSCs by StaMPS from the comparison in Figure 7b,e. It was observed that more lounge bridges pixels on the left side (see the *L* region in Figure 3), most of which are overlaid by the roof scatter and ground scatter, were selected by StaMPS. For the last step, more adjacent PS pixels were kept by IStaMPS than by StaMPS from the comparison of Figure 7c,f. Stability of the final selected PS points was demonstrated from subsequent retrieving results.

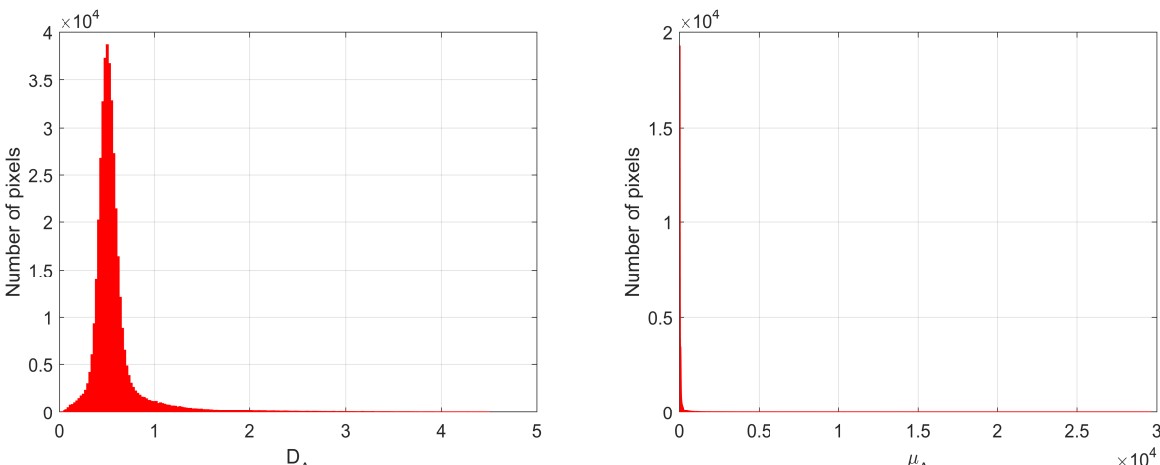

**Figure 5.** The distribution histograms of $D_A$ and $\mu_A$ of the dataset in PS selection of Step 1.

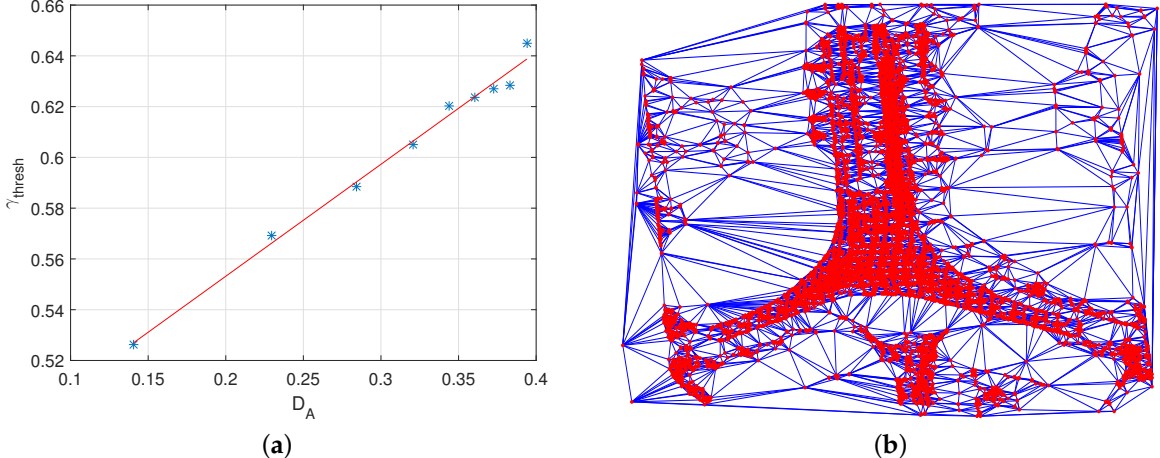

**Figure 6.** PS selection processes of Step 2 and 3: (**a**) the linearly fitted model between spatial coherence threshold and $D_A$; and (**b**) the Delaunay triangulation network of the updated PSCs.

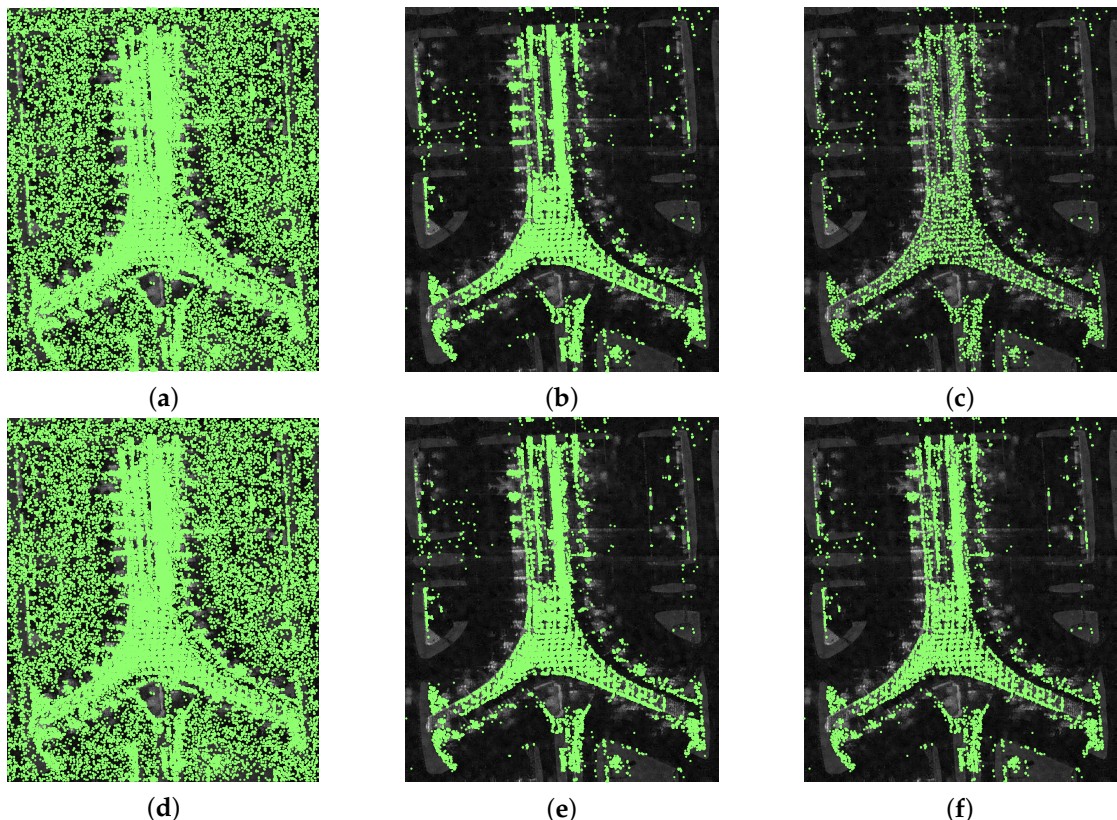

**Figure 7.** Comparison of PSCs results of StaMPS and IStaMPS (green points on the SAR image represent the PSCs): StaMPS PSCs after (**a**) Step 1, (**b**) Step 2, and (**c**) Step 3; and IStaMPS PSCs after (**d**) Step 1, (**e**) Step 2, and (**f**) Step 3.

*4.2. Topography and Deformation Rate Comparisons*

4.2.1. T3 E Site

After selection of the measurement points, the residual topography and deformation rate in the radar LOS direction of measurement points were retrieved by StaMPS and IStaMPS. Their maps of reconstructed height and mean deformation rate are illustrated in Figures 8 and 9.

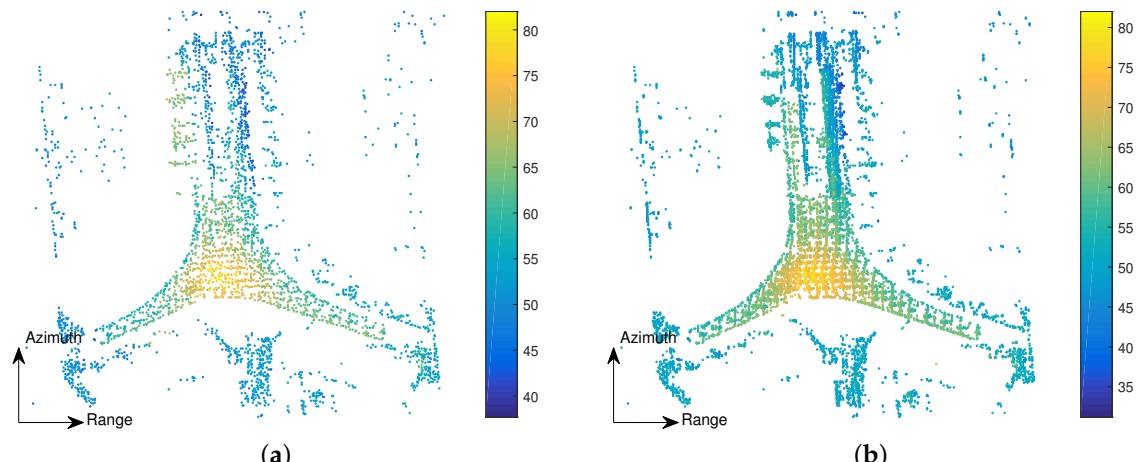

**Figure 8.** The reconstructed height maps of PS in T3 E area by: (**a**) StaMPS; and (**b**) IStaMPS [m].

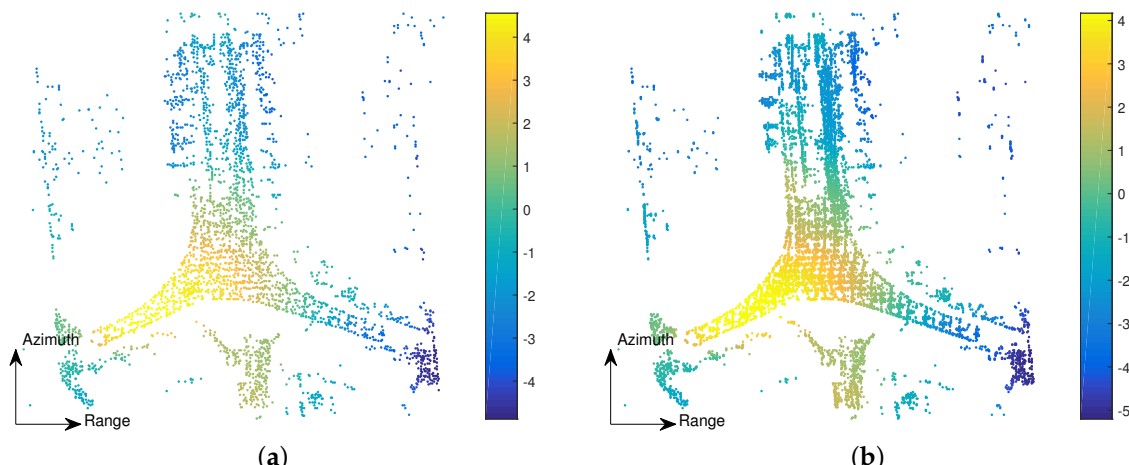

**Figure 9.** The mean deformation rate maps of PS in T3 E area by: (**a**) StaMPS; and (**b**) IStaMPS ([mm/year]).

The overall height maps after adding back the coarse topography for both StaMPS and IStaMPS are shown in Figure 8. Since the airport lies in a flat area with an average altitude of about 35 m [47,48], and the maximum height of T3 E relative to the flat ground is about 47 m [49], the maximum value of absolute height is about 82 m. Although the realistic streamlined shape at top of the building was reconstructed by these two methods, the proposed IStaMPS provides a much better height map in the most likely layover region $L$. Besides, the height of left lounge bridges by StaMPS goes against the truth in the $L$ region. Accordingly, IStaMPS has shown a significant improvement in measuring topography, especially for layover areas.

As shown in Figure 9, the deformation contour retrieved by IStaMPS is in good agreement with that by StaMPS. Moreover, the mean deformation rate of IStaMPS is in accord with the historic data in [37,47,48], demonstrating the effectiveness of IStaMPS in retrieving deformation. Although the two results have similar deformation patterns, the proposed strategy provides much denser measurements in building regions.

The SAR acquisition geometry along the $S$ profile in range and elevation plane marked in the top-left sub-image of Figure 3 is shown in Figure 10, where both sides of lounge bridges and symmetric main structure of T3 E building can be observed. Due to the side-looking geometry, the layover area most likely happens in the leftmost lounge bridge $L_l$, and shadow area most likely happens in the leftmost lounge bridge $L_r$. These areas are challenging to the PS-InSAR technique. It is worth noting that the rightmost lounge bridge is slightly shorter than the left one. Thus, the regions of lounge bridges $A_1$, $A_2$, $A_3$, $A_4$, $A_5$ and $A_6$ were selected, whose roofs are of the same height relative to ground, to assess the reliability of IStaMPS in most likely layover areas. The mean heights of the common PS points in each region were compared to show the accuracy of both methods. The height statistics of each region are illustrated in Figure 11. It was observed that the mean heights of all regions were approximately equal for IStaMPS, while there was large bias between the left and the right for StaMPS. Thus, it can be seen that IStaMPS is more reliable than StaMPS. Because each region contains scattering from the roof and facade of lounge bridges, there exist some differences between the mean height of five regions. Accordingly, the std of each region is not zero in Figure 11.

Subsequently, the advantage of IStaMPS over StaMPS in monitoring deformation is demonstrated in the following two aspects. On the one hand, the accuracy of retrieving height directly indicates the accuracy of mean deformation rate and thus the improved performance of IStaMPS over StaMPS in height reconstruction can be demonstrated by its improved performance in retrieving the mean deformation rate. On the other hand, the statistical results for the same region $B$ marked with rectangle in Figure 3 were compared. The spatial statistical results of 25 observations are shown in Figure 12. The maximum stds for the first and the last observations are 4.2035 mm/year and 4.5798 mm/year for StaMPS, and 3.5094 mm/year and 3.8672 mm/year for the proposed IStaMPS. Since the maximum std

indicates uncorrelation of temporal deformation, the improved results by IStaMPS over StaMPS in retrieving deformation can also be seen from the statistical results.

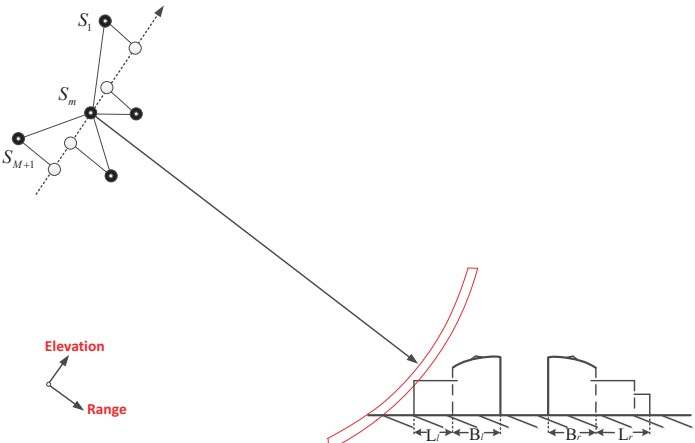

**Figure 10.** SAR acquisition geometry along the *S* profile in range and elevation plane marked in top-left sub-image of Figure 3. $L_l$ and $L_r$ represent lounge bridges of both sides, and $B_l$ and $B_r$ represent the symmetric main structure of the T3 E building.

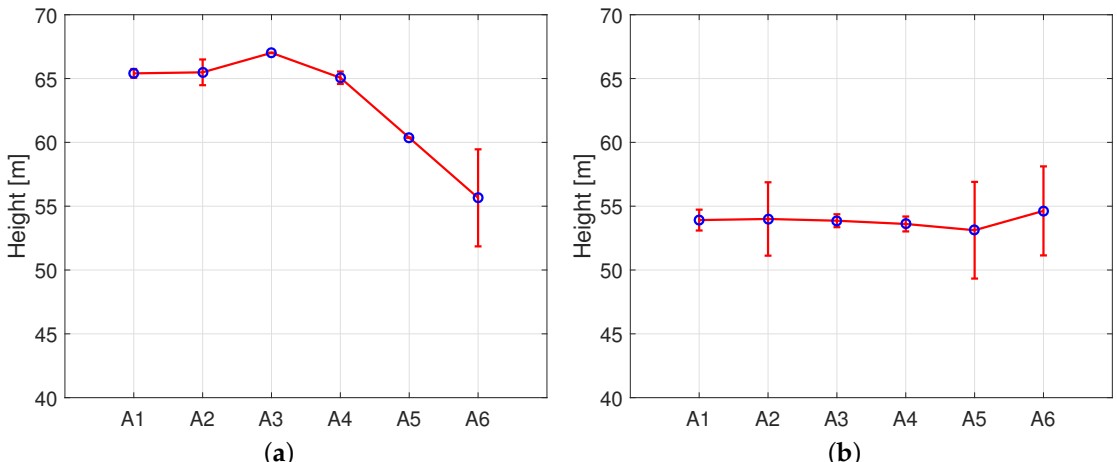

**Figure 11.** The mean height scatterplot with std bars of bridge roofs $A_1$, $A_2$, $A_3$, $A_4$, $A_5$, and $A_6$ marked in Figure 3: (**a**) StaMPS; and (**b**) IStaMPS.

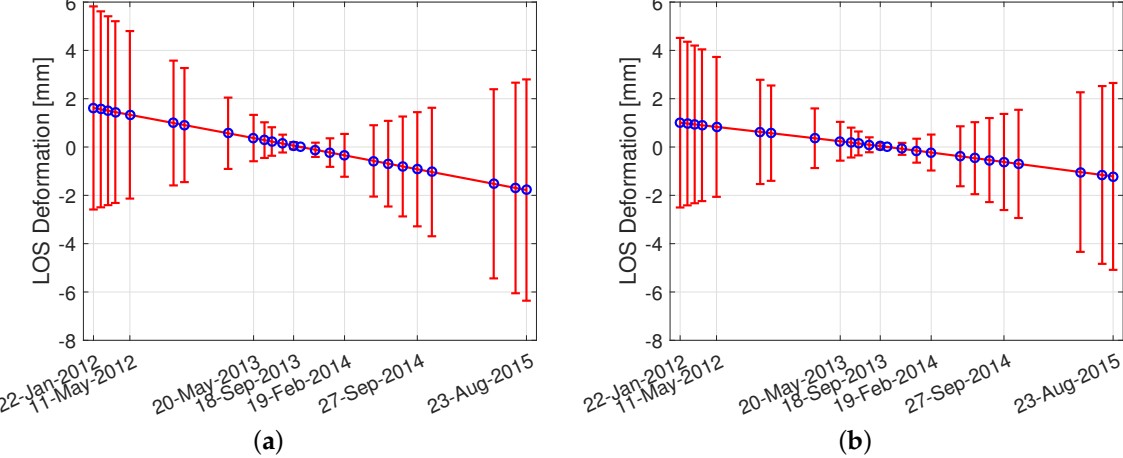

**Figure 12.** Scatterplot with std bars of retrieved deformation rates for (**a**) StaMPS and (**b**) IStaMPS.

### 4.2.2. Chaobai River

Due to the aforementioned PS selection process for T3 E area, here the reconstructed height and mean deformation rate maps retrieved by both methods were compared directly to discuss the performance of IStaMPS for non-urban areas. As shown in Figure 13, similar results were obtained by both methods in the natural area, while the proposed IStaMPS provided denser measurement points for all the man-made or natural targets on both sides of the river. Quantitatively, a total of 14,681 PS pixels and 37,771 PS pixels over the Chaobai River site were detected by StaMPS and IStaMPS, respectively. The density of the reliable measurement points detected by IStaMPS is about 2.6 times that by StaMPS. Since the pixels scattering from the river bridge share approximately equal height and deformation, the statistics of common points detected by both methods corresponding to region *R* of Figure 3 were compared quantitatively, as mentioned previously. The approximately equal mean height and deformation were obtained. However, the height std of StaMPS is 1.12 m, which is higher than the 0.99 m of the proposed IStaMPS. The maximum stds of the first and the last observations are 0.31 mm/year and 0.27 mm/year for StaMPS, and 0.36 mm/year and 0.38 mm/year for the proposed IStaMPS. These results clearly demonstrate the advantage of IStaMPS over StaMPS for non-urban areas.

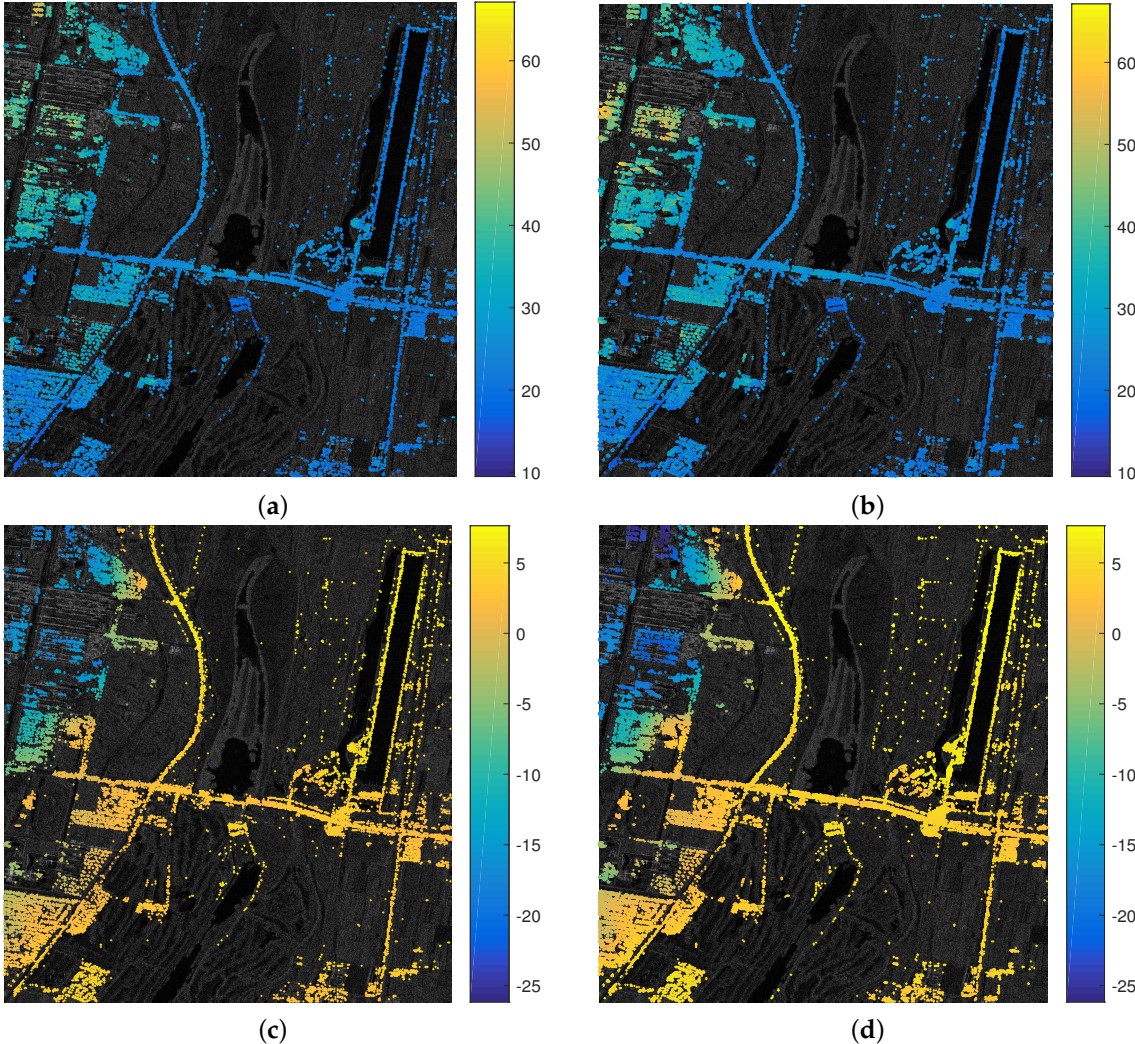

**Figure 13.** The reconstructed height and mean deformation rate maps of PS in Chaobai River area. The amplitude is plotted in the background. Height maps retrieved by (**a**) StaMPS and (**c**) IStaMPS ([m]); and mean deformation rate maps retrieved by (**b**) StaMPS and (**d**) IStaMPS ([mm/year]).

### *4.3. Deformation Time Series of the T3 E Building*

A more detailed analysis from the IStaMPS results on the spatiotemporal characterization of land subsidence/uplift is discussed in this subsection. The acquisition on 22 January 2012 is set as the reference image, and the spatial-temporal subsidence and uplift evolution are clearly visible over time. As illustrated by cumulative deformation maps in Figure 14, LOS deformation time series over the T3E site identified by IStaMPS ranged from −46.84 mm to 40.32 mm during 2012–2015 for the TerraSAR-X datasets. The region close to the runways experienced a slow subsidence, while the northwest region of the T3 E building experienced a land uplift. This result is in agreement with previous studies in [37,47,48]. The maximum linear uplift rate is about 4.2 mm/year over the whole monitoring period (see Figure 9).

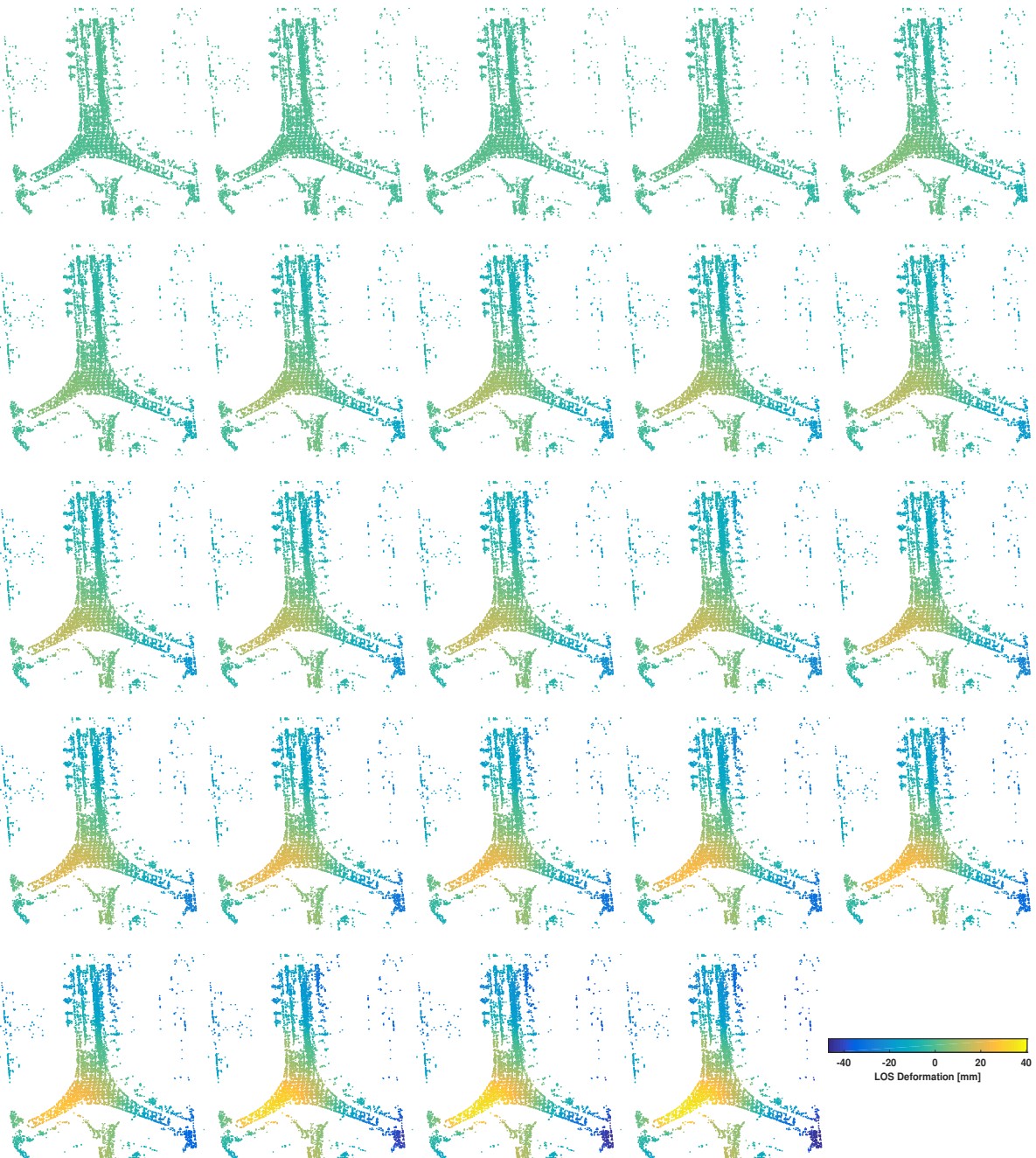

**Figure 14.** The LOS deformation time series over T3 E retrieved by the IStaMPS. All of the cumulative deformation maps are relative to the first image on 22 January 2012 and shown in SAR coordinate.

## 5. Conclusions

In this study, an improved PS-InSAR method, which can avoid the special overlaid pixels within multiple strong scatters for complex scenes and effectively remove the unstable adjacent pixels, was proposed based on the widely adopted method StaMPS. The PS pixels are selected with high density by the improved PS selection strategy. Moreover, the topography information not provided in StaMPS can be accurately measured in IStaMPS. A total of 25 TerraSAR-X/TanDEM-X Stripmap images acquired from June 2012 to August 2015 were used to test their performances. Although both methods have provided similar results in most regions, there are some differences, mainly concerning PS density and retrieving results in layover regions. The number of detected PS points is increased by 1.8 and 1.6 times for the T3 E building of the Beijing Capital International Airport and the Chaobai River area of Beijing Shunyi District, respectively. Moreover, the proposed IStaMPS achieved better results in topography measuring and deformation monitoring for complex buildings and natural scenes. Finally, the spatial-temporal evolution in the T3 E site was investigated with IStaMPS, and an exceptional uplift zone was detected in the northwest region of the T3 E building, with a maximum uplift rate of 4.2 mm/year.

**Author Contributions:** Conceptualization, H.X. and B.Y.; formal analysis, B.Y.; funding acquisition, H.X., C.L. and J.L.; investigation, B.Y. and J.G.; methodology, H.X. and B.Y.; resources, C.L. and J.L.; supervision, J.G., C.L. and J.L.; validation, B.Y., W.L. and J.G.; and writing—original draft, B.Y. and W.L.

**Funding:** This work was supported by the National Natural Science Foundation of China under Grant No. 61471020.

**Acknowledgments:** We particularly thank the National Aeronautics and Space Administration (NASA) for making SRTM DEM data available. Moreover, the provision of StaMPS by Stanford University is also gratefully acknowledged.

**Conflicts of Interest:** The authors declare no conflict of interest.

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
