# Peer review of "An Improved Stanford Method for Persistent Scatterers Applied to 3D Building Reconstruction and Monitoring"

_remotesensing, doi:10.3390/rs11151807_

Round 1

Reviewer 1 Report

Dear authors,

some minor changes are needed. Check the general English editing, sometimes the word choice is not correct.

Please find attached the reviewed manuscript.

Author Response

Thanks for your comments. We have revised the manuscript according to your comments listed in Response to Reviewer #1. In addition, the estimations of topography and deformation have been improved in the revised manuscript.

Reviewer 2 Report

In the paper titled "An Improved Stanford Method for Persistent Scatterers Applied to 3D Building Reconstruction and Monitoring" is proposed an improvement on the Stanford Method for Persistent Scatterers, which is mainly concentrated on the PS selection over a set of radar images. PSI is one of the most sophisticated technique for sistematic analysing tiny positional variations over a specific region. Besides, it is also one of the most expensive, due to the fact it requires not only one single campaing, but the images has to be taken constantly (25 images taken for this study). It all make this approach a bit challenge, and gives this paper great impact. 

Said that, the paper present a good and clear perspective surrounding the topic. The references to known methods were correctly made, always taking care to cite not one but multiple alternatives of other methods and what they are intending to do. Another positive point is the  english, which I barely suggested changes. The improvement on StaMPS is well presented, with a fluxogram, explanations about each step, the objective was directly point out, etc. 

I would suggest more detailed images, to understant a bit more the process of PS selection and the influences of radar acquisition geometry in the analysis.

------------------------------------------------------------

Abstract: The abstract is OK. Very good written and well structured. No changes needed. 

------------------------------------------------------------

Introduction

P. 1 - Line 3: Check the use of slashes. Use commas: [...and anthropogenic subsidence or uplifts, due to fluid...]

P. 1 - Line 4: Check the use of lowercase letters for methods or well-known procedures, such as "synthetic aperture radar (SAR)". Please, use style: "Synthetic Aperture Radar (SAR)"

------------------------------------------------------------

Methodology

The related work is incorporated in this section, which is not bad if you know how to insert and where. 

The method proposed was presented here, with clear explanations, references, with one section for each of modeling stage. Good.

No concerns about methodology.

------------------------------------------------------------

Results

P. 1 - Line 3: "...information.[SPACE]All results "

The figures does not speak for itself. Please, give a more complete description in Figures 7, 8, and 9 (improve the captions). Figure 7, which row is IStaMPS and StaMPS? What could be notice with them? Are they different?

------------------------------------------------------------

Conclusions

The conclusions are relevant and supports the results presented. I would suggest to include more comments about the method proposed, i.e. the contributions. 

------------------------------------------------------------

References

Mr. Ferreti, Hooper, and others are references usually among PSI analysis. See them on the list means, at least, a good study involving these topics. 

Author Response

Thanks for your comments. We have revised the manuscript according to your comments listed in Response to Reviewer #2. In addition, the estimations of topography and deformation have been improved in the revised manuscript.

Reviewer 3 Report

This paper modified the existing StaMPS PSInSAR method. The idea is generally straightforward and the results look promising. However, there are several critical issues that need more justifications. 

I would suggest the authors add more new InSAR references in the introduction, such as: 

a)A review of ten-year advances of multi-baseline SAR interferometry using TerraSAR-X data. Authors Xiao Zhu, Yuanyuan Wang, Sina Montazeri, Nan Ge. A detailed review of multi-baseline InSAR is given. b) A phase-decomposition-based PSInSAR processing method N Cao, H Lee, HC Jung IEEE Transactions on Geoscience and Remote Sensing 54 (2), 1074-1090. Both this and CAESAR are proposed to account for layover effects.

The main improvement of the proposed method is the capability to avoid the layover pixels by thresholding on ADI and MA. It would be more convincing if the authors can identify the layover pixels in the results. In Figure 7 to Figure 9, the authors claimed that the proposed method got better performance due to avoiding the layover pixels (e.g. bridges). However, I am not totally convinced by such claims. It would be better if the authors can isolate the layover pixels that were used in the StaMPS.

Figure 10 shows the height of the bridge roofs, which show that the proposed method gets better results. However, I can see that there are more than 5 bridges. I would ask the authors to identify all the other bridges and do a comprehensive comparison instead of "selecting" the good ones. 

Please change the x limit of Figure 5 (b). 

What's the performance of the proposed method in none-urban area or urban area with less density?

Author Response

Thanks for your supportive comments. We have revised the manuscript according to your comments listed in Response to Reviewer #3. In addition, the estimations of topography and deformation have been improved in the revised manuscript.

Round 2

Reviewer 3 Report

The authors answered all of my questions. 

This manuscript is a resubmission of an earlier submission. The following is a list of the peer review reports and author responses from that submission.